# HUMAN MOTION DIFFUSION AS A GENERATIVE PRIOR

**Yonatan Shafir**,[*] **Guy Tevet**,[*] **Roy Kapon and Amit H. Bermano**
Tel Aviv University, Israel
{Shafir2, guytevet}@mail.tau.ac.il

## ABSTRACT

Recent work has demonstrated the significant potential of denoising diffusion models for generating human motion, including text-to-motion capabilities. However, these methods are restricted by the paucity of annotated motion data, a focus on single-person motions, and a lack of detailed control. In this paper, we introduce three forms of composition based on diffusion priors: sequential, parallel, and model composition. Using sequential composition, we tackle the challenge of long sequence generation. We introduce DoubleTake, an inference-time method with which we generate long animations consisting of sequences of prompted intervals and their transitions, using a prior trained only for short clips. Using parallel composition, we show promising steps toward two-person generation. Beginning with two fixed priors and a few two-person training examples, we learn a slim communication block, ComMDM, to coordinate interaction between the two resulting motions. Lastly, using model composition, we first train individual priors to complete motions that realize a prescribed motion for a given joint. We then introduce DiffusionBlending, an interpolation mechanism to effectively blend several such models to enable flexible and efficient fine-grained joint and trajectory-level control and editing. We evaluate the composition methods using an off-the-shelf motion diffusion model, and further compare the results to dedicated models trained for these specific tasks. [1]

## 1 INTRODUCTION

Human Motion Generation has recently experienced a tremendous leap forward. The recent elaborate language models (Radford et al., 2021; Devlin et al., 2019) and diffusion generation approach (Sohl-Dickstein et al., 2015; Ho et al., 2020) have quickly found their way into the field, yielding motion generation models that produce diverse and high-quality sequences from text or other forms of control (Tevet et al., 2023; 2022; Petrovich et al., 2022; Guo et al., 2022). In turn, these models have been already applied in the world of gaming, and hold the potential to open the field of character animation to novices and professionals alike.

However, the main problem the field of human motion generation has always struggled with and is still struggling with is data. Motion data is typically either acquired by elaborate motion capture settings (Joo et al., 2015) or crafted by artists (Adobe Systems Inc., 2021). Both cases eventually lead to expensive and relatively small and homogeneous datasets (Punnakkal et al., 2021; Guo et al., 2022). For example, the datasets that current models are trained on, consist almost exclusively of short, single-person sequences. In the absence of data, tasks like multi-person interaction and long sequence generation are left behind, with poor generation quality.

In this paper, we show that pretrained diffusion-based motion generation models can be leveraged as priors for composition, allowing out-of-domain motion generation and efficient control. Contrary to the high data consumption reputation of diffusion models, we show three methods that overcome the cost barrier using the aforementioned prior, enabling non-trivial tasks in few-shot or even zero-shot settings.

In particular, we choose a pretrained Motion Diffusion Model (MDM) (Tevet et al., 2023) to serve as the prior. MDM achieves state-of-the-art results in the text-to-motion and action-to-motion tasks for short single-person sequences, and has already been demonstrated to generalize well to condi-

---

[*]Equal contribution
[1]Our code and trained models are available at https://github.com/priorMDM/priorMDM.

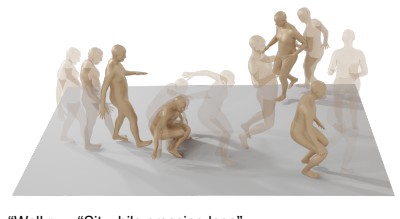 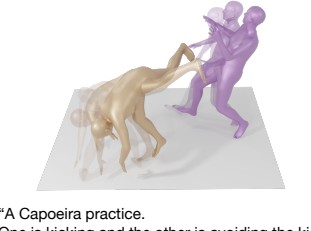 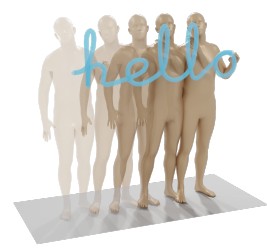

"Walk" → "Sit while crossing legs" → →"Long leap forward" →"Running in a circle"

"A Capoeira practice. One is kicking and the other is avoiding the kick."

Figure 1: We suggest three novel motion composition methods, all based on the recent Motion Diffusion Model (MDM). **(Left) Sequential composition** generating an arbitrary long motion with text control over each time interval. **(Middle) Parallel composition** generating two-person motion from text. A different color represents a different person - both are generated simultaneously given the text prompt. **(Right) Model composition** achieving accurate and flexible control by blending models with different control signals - here writing "hello" in mid-air.

tions from other domains (Tseng et al., 2022), and to corrections performed between the sampling iterations (Yuan et al., 2022).

Using this prior, we demonstrate three forms of composition:

**(1) Sequential composition**, where short sequences are concatenated to create a single long and coherent motion; **(2) parallel composition**, where two single motions are coordinated to perform together; and **(3) model composition**, where the motions generated by models with different control capabilities are blended together for composite control.

Our DoubleTake method (Figure 1-Left), suggests a *sequential composition* by carefully composing two generated motions in time, including the transition between them, and enables the efficient generation of long motion sequences in a zero-shot manner. Using it, we demonstrate 10-minute long fluent motions that were generated using a model that was trained only on up to 10 seconds long sequences (Guo et al., 2022; Punnakkal et al., 2021). In addition, due to the composite nature of the generation, DoubleTake allows individual control for each motion interval, while maintaining consistent motion and transitions. This result is fairly surprising considering that such transitions were not explicitly annotated in the training data. DoubleTake consists of two phases for every diffusion iteration - in the first step, the individual motions, or intervals, are generated together in the same batch, each aware of the context of its neighboring intervals. Then, the second take refines the transitions between intervals to better match those generated in the previous phase.

For *parallel composition*, we consider a few-shot setting, and enable textually driven two-person motion generation for the first time (Figure 1-Middle). Using our prior-based approach, we demonstrate promising two-person motion generation using only as few as a dozen training examples. The key idea is that in order to learn human interactions, we only need to enable prior models to communicate with each other throughout the diffusion process. Hence, we learn a slim communication block, ComMDM, that passes a communication signal between the two frozen priors through intermediate activation maps.

Finally, we introduce a novel control mechanism via *model composition*. We observe that the motion inpainting process suggested by Tevet et al. (2023) does not extend well to more elaborate yet important motion tasks such as trajectory and end-effector tracking. Hence, we first show that fine-tuning the prior for this task yields satisfying results while controlling even just a single end-effector. Then, we introduce the DiffusionBlending technique, which generalizes classifier-free guidance (Ho & Salimans, 2022) to compose together different fine-tuned models and thus enables cross combinations of keypoints control on the generated motion. This enables surgical and flexible control for human motion that comprises a key capability for any animation system (Figure 1-Right).

We demonstrate, both quantitatively and qualitatively, that these inexpensive composition methods extend a more elaborately trained motion prior and outperform dedicated previous art in the respective tasks (Wang et al., 2021; Athanasiou et al., 2022).

## 2 RELATED WORK

**Motion Diffusion Models.** Very recently, MDM (Tevet et al., 2023), MotionDiffuse (Zhang et al., 2022), MoFusion (Dabral et al., 2023), and FLAME (Kim et al., 2022) successfully implemented motion generation neural models using the Denoising Diffusion Probabilistic Models (DDPM) (Ho et al., 2020) setting, which was originally suggested for image generation. MDM enables both high-quality generation and generic conditioning that together comprise a good baseline for new motion generation tasks. EDGE (Tseng et al., 2022) followed MDM by extending it for the music-to-motion task. SinMDM (Raab et al., 2023) adapted MDM to non-human motions using a single-sample learning scheme. PhysDiff (Yuan et al., 2022) added to MDM a pre-trained physical model based on reinforcement learning which enforces physical constraints during the sampling process. These examples demonstrate the flexibility of MDM to novel tasks. In line with our motion control context, Jiang et al. (2022), Du et al. (2023) and Castillo et al. (2023) reconstruct full-body motion from a headset and hand controllers for Virtual Reality applications. While they focus on predicting the original motion, our approach models motion distribution based on one or a few joints.

**Long-Sequence Motion Generation.** Motion Graphs (Kovar et al., 2008) can synthesize long motions via traversing discrete poses given a data corpus. This approach is limited to existing data and will fail to generalize for elaborate textual conditions. RNN-based motion generation tends to collapse into constant poses. Martinez et al. (2017) and Zhou et al. (2018) overcome this issue by feeding the model with its own generated frames during training for the task of prefix completion. Yet, those methods are still limited to the relatively short sequences of the available data. More recently, several works suggested breaking the data limitation by auto-regressively generating short sequences each one conditioned on a textual prompt and the suffix of the previous sequence. Transitions were either learned according to a smoothness prior (Athanasiou et al., 2022; Mao et al., 2022) or from data (Athanasiou et al., 2022; Wang et al., 2022), using the BABEL dataset (Punnakkal et al., 2021), which explicitly annotates transitions between actions. EDGE (Tseng et al., 2022) suggested the unfolding method to generate long sequences with SLERP interpolating between every two neighboring sequences. Contrarily, our DoubleTake suggests an unfolding method that leverages diffusion and blends the motion together at each denoising step. More recently, DiffCollage (Zhang et al., 2023) suggested a diffusion-based solution for the task, utilizing a factor graph representation arranged as a linear chain.

**Multi-Person Motion Generation.** Data scarcity is a major obstacle for multi-person motion generation, and the number of works is limited accordingly. MuPoTS-3D dataset (Mehta et al., 2018) includes 20 real-world multi-person sequences; CMU-Mocap (CMU) and 3DPW (Von Marcard et al., 2018) includes 55 and 27 two-person motion sequences respectively. Yin et al. (2018) suggested overcoming the data barrier by exploiting 2D information. Recently, Song et al. (2022) contributed the synthetic multi-person GTA Combat dataset. None of the datasets is textually (or otherwise) annotated, hence, the recent MRT (Wang et al., 2021) and SoMoFormer (Vendrow et al., 2022) models learned the unsupervised prefix completion task. Both learned motions under the DCT transform, which promotes smoothness and unrealistic motion, although improving L2 error measures. Recently, DuMMF (Xu et al., 2022) presented a stochastic approach for multi-person motion completion. Concurrent to this work, InterGen (Liang et al., 2023) follow up and extend the two instances' communication principle presented in ComMDM and conducted a new benchmark.

**Human Motion Priors.** VPoser (Pavlakos et al., 2019) is a human pose auto-encoder, trained on the AMASS motion capture dataset (Mahmood et al., 2019). It is used as a prior for motion applications, such as motion denoising, fitting SMPL (Loper et al., 2015) model to joint location and as a pose code book for motion generation (Hong et al., 2022). More recently, Tiwari et al. (2022) showed that such prior can be learned as an implicit model. MoDi (Raab et al., 2022) is an unsupervised motion generator, adapted from StyleGAN (Karras et al., 2019). Without further training, it enables latent space editing and motion interpolation. Contrary to those examples, MotionCLIP (Tevet et al., 2022) uses priors from the image and text domains to learn motion. It aligns the motion manifold with CLIP (Radford et al., 2021) latent space. This enables inheriting the knowledge learned by CLIP to generate motions out of the data limitations. In the diffusion context, MDM adapts diffusion image inpainting (Song et al., 2020; Saharia et al., 2022) for motion editing applications. In this work, we extend this principle by solving non-trivial motion tasks in few to zero-shot settings. More recently, MLD (Xin et al., 2022) learned a latent diffusion model, similar to LDM (Rombach et al., 2022b),which enables generating motion latent code instead of the motion itself, and lets a larger and pre-trained motion generator translate it into the physical space.

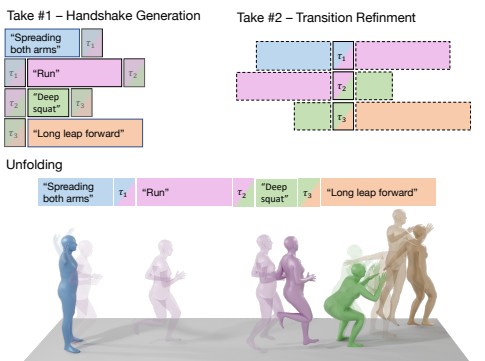

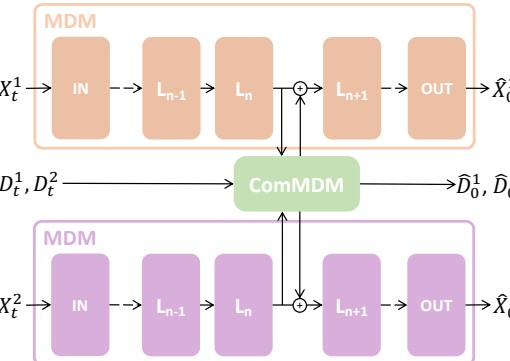

Figure 3: **DoubleTake overview.** We generate arbitrarily-long sequences with text and length control per interval using a fixed motion diffusion prior. At the **first take**, we generate each interval as a single sample handshaking neighboring samples. At each denoising iteration, the handshakes are forced to be equal to eventually compose one long sequence. To refine the transition between intervals, the **second take** partially noise the handshakes and clean them conditioned on the neighboring intervals using a soft mask. Solid frames mark generation or refinement; Dashed frames mark input motion to the take.

Figure 4: **ComMDM overview.** Using two fixed MDM models, we train a slim communication block (ComMDM) for two-person motion generation. ComMDM gets as input the activations of transformer layer $L_n$ from both actors and outputs a correction term which is added to the same activations. Optionally, ComMDM also predicts the initial poses $D^i$ of the two persons. IN and OUT stand for the linear input and output layers of the transformer.

## 3 METHOD

In this work, we use the recent Motion Diffusion Model (MDM) (Tevet et al., 2023), pre-trained for the task of text-to-motion, to learn new generative tasks. We represent Human Motion as a sequence of poses $X = \{x^i\}_{i=1}^N$ where $x^i \in \mathbb{R}^D$ represent a single pose. Specifically, we use the SMPL (Loper et al., 2015) representation for experiments with the BABEL (Punnakkal et al., 2021) dataset, including joint rotations and global positions on top of a single human identity ($\beta = 0$). For all other experiments, we use the HumanML3D (Guo et al., 2022) representation, composed of joint positions, rotations, velocities, and foot contact information. MDM is a denoising diffusion model based on the DDPM (Ho et al., 2020) framework. It assumes $T$ noising steps modeled by the stochastic process

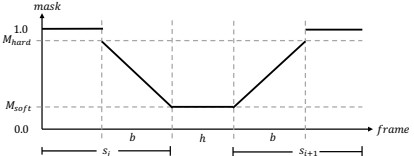

Figure 2: **Soft blending overview.** We allow **b** frames long linear masking between $M_{hard}$ to $M_{soft}$ such that during the **Second take** at every denoising step part of the originally generated motion (suffix or prefix) going through refinement to fit the transition.

$$q(X_t|X_{t-1}) = \mathcal{N}(\sqrt{\alpha_t}X_{t-1}, (1 - \alpha_t)I), \tag{1}$$

for a noising step $t \in T$, were $X_T \sim \mathcal{N}(0, I)$ is assumed. MDM models the denoising process: it predicts the clean motion $\hat{X}_0$ given a noised motion $X_t$, a noise step $t$ and a textual condition encoded to CLIP (Radford et al., 2021) space and represented by $c$. The model is learned with the standard $\mathcal{L}_{\text{simple}} = E_{X_0 \sim q(X_0|c), t \sim [1,T]}[\|X_0 - MDM(X_t, t, c)\|_2^2]$ together with geometric losses that regulate the joint position, velocity and foot contact. Sampling a novel motion from MDM is done in an iterative manner, according to Ho et al. (2020). In every time step $t$ the clean sample $\hat{X}_0$ is predicted and noised back to $X_{t-1}$. This is repeated from $t = T$ until $X_0$ is achieved.

In this Section, we present *sequential composition* with the DoubleTake method (3.1), which generalizes MDM to generate motions of arbitrary length without further training, through sequential composition. Then, we present *parallel composition* by employing a slim communication layer,

ComMDM (3.2), trained with as few as 10 interaction samples, for generating two-person motion. Lastly, we fine-tune MDM to control specific joints and present our *model composition* method, DiffusionBlending (3.3), that generalizes the classifier-free approach (Ho & Salimans, 2022) to achieve fine-grained control over the body with any cross combination of joints to be controlled.

## 3.1 LONG SEQUENCES GENERATION

Our goal is to generate arbitrarily long motions, such that each time interval of the motion is potentially controlled with a different text prompt and a different sequence length. We want the transitions between intervals to be realistic and to semantically match the neighboring intervals. Since available datasets are limited in motion length and often do not explicitly include transitions, we suggest approaching this task in a zero-shot manner, using a fixed generative prior that was trained with such short sequences.

We present DoubleTake (Figure 3), a two-stage inference-time process that suggests a parallel solution and generates the long motion in a single batch. Typically, approaches that were designed specifically for this task (Athanasiou et al., 2022; Mao et al., 2022; Wang et al., 2022) generate each such interval conditioned on the fixed suffix of the previous interval. In contrast, DoubleTake generates a prompted interval while observing both the previous and next intervals, which are generated simultaneously. In the first take, we generate each interval as a different sample in the denoised batch, such that each one is conditioned on its own text prompt and maintains a *handshake* with its neighboring intervals through the denoising process. Handshake, $\tau$, is defined as a short (about a second long) prefix or suffix of the motion, such that the prefix of the current motion is forced to be equal to the suffix of the previous motion. Each interval maintains two such handshakes as demonstrated in Figure 3. The handshake is maintained by simply overriding $\tau$ with the frame-wise average of the relevant suffix and prefix at each denoising step. This allows our model to generate long sequences that depend on the past and future motions while being aware of the whole sequence during the generation of each interval. The handshake length $h = |\tau|$ can be arbitrarily defined by the user, also on a per-transition level. However, in practice, we find that the choice of one-second-long handshakes is robust throughout our experiments. Formally, handshakes are forced to be equal at the end of each denoising iteration as follows:

$$\tau_i = (1 - \vec{\alpha}) \odot S_{i-1}[-h :] + \vec{\alpha} \odot S_i[: h] \tag{2}$$

where $S_i$ indicates the $i^{th}$ sequence $\alpha_j = j/h, \forall j : j \in [0 : h)$ and $\odot$ indicates a element-wise multiplication.

Looking at the generated handshaked motion however, we observe visually displeasing results, as artifacts and inconsistencies occur in the transitions between semantically different motions (i.e. "Run" and then "Crawl"). Consequently we suggest adding the *second take*, applied on the output of the first take. In the second take, we reshape our batch as shown in Figure 3, such that in each sample we get the transition sandwich $(S_i, \tau_i, S_{i+1})$. Now, we partially noise the sandwich $T'$ noising steps and denoise it back to $t = 0$ under our suggested *soft-masking* feature to refine transitions: In a regular inpainting mask, the content is either taken completely from the input, or is completely generated. We suggest a soft inpainting scheme, where each frame is assigned a soft mask value between $0$ and $1$ that dictates the amount of refinement the second take performs on top of the first take's result. To this end, we define the masks $M_{soft}$, $M_{hard}$ for the interval $S$ and hanshake $\tau$ respectively, with a short, $b$ frames long, linear transition between the mask values as demonstrated in Figure 2. Finally, we construct the long sequence by *unfolding it*, i.e. by reshaping each sequence and transition back to its linear place as demonstrated in Figure 3 bottom.

## 3.2 TWO-PERSON GENERATION

Our goal is to simultaneously generate motion of two people interacting with each other. The limited data availability dictates a few-shot learning solution. Our key insight is that by dedicating a fixed generator for each person in the scene the motion remains in the human motion distribution, and we only need to learn to coordinate between the two. Hence, we introduce ComMDM (Figure 4), a single-layer transformer model that is trained to coordinate between two instances of a fixed MDM (one for each person). ComMDM is placed after transformer layer $n$, gets as inputs the output activations of this layer from both models $(O_t^{1,(n)}, O_t^{2,(n)})$ and outputs a correction term for each of the two models $\Delta O_t^{i,(n)}$. To further reduce the number of learned parameters, we exploit symmetry

considerations and output only one correction term, then the output to be corrected is entered first, such that the corrected output is $\tilde{O}_t^{i,(n)} = O_t^{i,(n)} + ComMDM(O_t^{i,(n)}, O_t^{3-i,(n)})$. We note that in some datasets, such as HumanML3D, all motions are processed to start with the root at the origin and facing the same direction. Hence, naively using ComMDM on a model that was trained with such data will result in two people both being placed at the origin at the beginning of the motion. To mitigate that, ComMDM additionally learns $D$, the initial pose of each person at the first frame as a part of the diffusion process. Hence, the full implementation of ComMDM is $\Delta O_t^{i,(n)}, \hat{D}_0^i = ComMDM(O_t^{i,(n)}, O_t^{3-i,(n)}, D_t^i, t)$.

We freeze the weights of the MDM instances and train only ComMDM with the $L_{\text{simple}}$ loss. We learn two motion tasks; For prefix completion, we use a fine-tuned version of MDM for prefix completion (See 3.3) and completely mask the textual condition. For the text-to-motion task, we use a regular instance of MDM and mask the textual condition with a probability of $10\%$ to support classifier-free guidance.

### 3.3 Fine-Tuned Motion Control

Our goal is to generate full-body motion controlled by a user-defined set of input features. These features can be root trajectory, a single joint, or any combination of them. We require a self-coherent generation that semantically adheres to the control signal. For instance, when specifying the root trajectory of a person to move backward, we expect the generated motion to have the legs adjusted to walking backward. As we show in Section 4.3, the motion in-painting method suggested by Tevet et al. (2023) fails to meet this requirement.

**Single Control Fine-Tuning.** Consequently, inspired by Rombach et al. (2022a), we introduce a fine-tuning process to yield a model that adheres to the control features. In essence, our method works by masking out the noise applied to the ground-truth features we wish to control, during the forward pass of the diffusion process. This means that during training, the ground-truth control features propagate to the input of the model, and thus, the model learns to rely on these features when trying to reconstruct the rest of the features. Algorithm 1 describes the fine-tuning process for trajectory control task. For sampling, we follow the core idea of the finetuning process: After we get the model's prediction of $x_0$, we inject the editing features into it. Then, in the forward process from the predicted $x_0$ to $x_{t-1}$, we mask out the noise in the control features to allow them to cleanly propagate into the model. Algorithm 2 defines this sampling process for trajectory control task. The fine-tuning stage requires less than $20K$ steps to generate visually pleasing results. It allows us to easily acquire a dedicated model for a given control task.

**DiffusionBlending.** A fine-tuned model for every possible control task is sub-optimal. Hence, we suggest DiffusionBlending, a *model composition* method for using multiple models for composite control tasks. For instance, if we wish to dictate both the trajectory of the character and its left hand, we can blend the model that was trained solely for trajectory control and the model that was trained only for the left hand.

To control cross combinations of the joints (i.e. both the root and the end effector as in Figure 1), we extend the core idea of the classifier-free approach (Ho & Salimans, 2022) and present DiffusionBlending. The classifier-free approach suggests interpolating or extrapolating between the conditioned model $G$ and the unconditioned model $G^{\emptyset}$. We argue that this idea can be generalized to any two "aligned" (see definition in (Wu et al., 2021)) diffusion models $G^a$ and $G^b$ that are conditioned on $c_a$ and $c_b$ respectively. Then sampling with two conditions simultaneously is implemented as $G_s^{a,b}(X_t, t, c_a, c_b) = G^a(X_t, t, c_a) + s \cdot (G^b(X_t, t, c_b) - G^a(X_t, t, c_a))$, with the scale parameter $s$ trading-off the significance of the two control signals.

For the general case of $N$ joints corresponding to $N$ fine-tuned models $\{G^n\}_{n=1}^N$ the Diffusion-Blending will be defined, $G_s^{[N]}(X_t, t, \{c_n\}_{n=1}^N) = \sum_{n=1}^N s_n \cdot G^a(X_t, t, c_n))$, where $\sum_{n=1}^N s_n = 1$.

## 4 Experiments

### 4.1 Long Sequences Generation

For long sequence generation with our DoubleTake method, we use a fixed MDM (Tevet et al., 2023) trained on the HumanML3D (Guo et al., 2022) dataset, originally trained with up to 10 seconds long

| | Motion | | | | Transition (70 frames) | | Transition (30 frames) | |
|---|---|---|---|---|---|---|---|---|
| | R-precision ↑ | FID↓ | Diversity→ | MultiModal-Dist↓ | FID↓ | Diversity→ | FID↓ | Diversity→ |
| Ground Truth | 0.62 | $0.4 \cdot 10^{-3}$ | 8.51 | 3.57 | $0.8 \cdot 10^{-3}$ | 8.23 | $0.9 \cdot 10^{-3}$ | 8.33 |
| TEACH | 0.46 | 1.12 | **8.28** | 7.14 | 3.86 | **7.62** | 7.93 | 6.53 |
| Double Take (ours) | 0.43 | 1.04 | 8.14 | 7.39 | **1.88** | 7.00 | **3.45** | 7.19 |
| + Trans. Emb | **0.48** | **0.79** | 8.16 | **6.97** | 3.43 | 6.78 | 7.23 | 6.41 |
| + Trans. Emb + geo losses | 0.45 | 0.91 | 8.16 | 7.09 | 2.39 | 7.18 | 6.05 | 6.57 |

Table 1: **Quantitative results on the BABEL test set.** All methods use the real motion length from the ground truth. '→' means results are better if the metric is closer to the real distribution. We run all the evaluations 10 times. Transition metrics were tested on two different lengths and it contains context from the suffix of the previous frame and the prefix of the next frame. tested on two different margin lengths since TEACH define transition of 8 frames. **Bold** indicates best result, _underline_ indicates second best result. R-precision reported is top-3.

motions. To compare with TEACH (Athanasiou et al., 2022), which was dedicatedly trained for this task, we train MDM for $1.25M$ steps on BABEL (Punnakkal et al., 2021), the same dataset TEACH was trained on with the same hyperparameters suggested by Tevet et al. (2023) on a single NVIDIA GeForce RTX 2080 Ti GPU. For both datasets, we applied DoubleTake with a one-second-long transition length, $T' = 700$, $M_{hard} = 0.85$, $M_{soft} = 0.1$ and $b = 10$.

In both cases, we evaluate the generation using the evaluators and metrics suggested by Guo et al. (2022). In short, they learn text and motion encoders for the HumanML3D dataset as evaluators that map motion and text to the same latent space, then apply a set of metrics on the generated motions as they are represented in this latent space. *R-precision* measures the proximity of the motion to the text it was conditioned on, *FID* measures the distance of the generated motion distribution to the ground truth distribution in latent space, *Diversity* measures the variance of generated motion in latent space, and *MultiModel distance* is the average $L2$ distance between the pairs of text and conditioned motion in latent space. For full details, we refer the reader to the original paper. We further conducted a user study comparing DoubleTake to TEACH. DoubleTake was preferred in about 82 to 85% of the times. For more details please refer to Appendix C.

Note that for the BABEL dataset, we trained the same evaluators following the setting defined by Guo et al. (2022). To provide a proper analysis, we generate a 32-intervals long sequence, then apply HumanML3D metrics on the intervals themselves, and once again for the transition. Note that the text-related metrics are not relevant for transitions.

Since the BABEL dataset annotates transitions as well, we suggest using our *Transition Embedding*: we choose to embed each frame with transition embedding signal, allowing the model better understand if the following frame belongs to transition or part of the motion. We then add this embedding to the frame's features. Additionally, we choose to train our model over the BABEL dataset with geometric losses as proposed in MDM. We note that whereas we do not apply any post-process to the motion, TEACH aligns the start of each interval to the end of the previous one and adds extra interpolation frames between the two. We observe that without this post-process TEACH produces poor transition, yet evaluated it with all the above to maintain fair conditions.

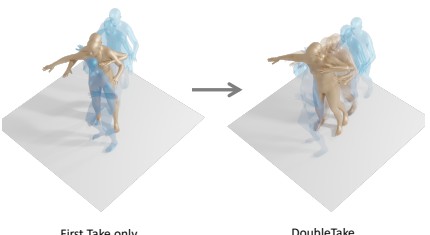

First Take only      DoubleTake

Figure 5: **DoubleTake transition refinement.** The second take refines the transitions generated in the first take to be more smooth and more realistic. Orange are subsequent transition frames and Blue are context intervals.

Table 1 presents quantitative results over the BABEL dataset, compared to TEACH. We evaluated the transitions with two variations - the first with fair margins from the intervals (70 frames) and the other with minimal possible margins for both DoubleTake and TEACH (30 frames which are 1 second). DoubleTake outperforms TEACH in terms of FID with all our methods. When considering transition evaluations the gaps in favor of DoubleTake are even larger. Figure 10 (Appendix) shows a qualitative comparison between the two approaches. Comparisons to DiffCollage (Zhang et al., 2023) were not performed since code is not available.

| | Motion | | | | Transition | |
|---|---|---|---|---|---|---|
| | R-precision↑ | FID↓ | Div.→ | M.-Dist↓ | FID↓ | Div.→ |
| Ground Truth | 0.80 | $1.6 \cdot 10^{-3}$ | 9.62 | 2.96 | 0.05 | 9.57 |
| DoubleTake (ours) | 0.59 | **0.60** | 9.50 | 5.61 | **1.48** | **8.90** |
| First take only | 0.59 | 1.00 | 9.46 | 5.63 | 2.15 | 8.73 |
| Second take only | 0.59 | 1.09 | 9.34 | **5.57** | 3.22 | 8.35 |
| DoubleTake ($b=0$) | 0.59 | 1.00 | 9.51 | 5.61 | 2.21 | 8.66 |
| DoubleTake ($b=20$) | 0.59 | 0.84 | 9.74 | 5.60 | 1.56 | 8.73 |
| DoubleTake ($h=30$) | 0.60 | 1.03 | 9.53 | 5.60 | 2.22 | 8.64 |
| DoubleTake ($h=40$) | 0.58 | 1.16 | **9.61** | 5.67 | 2.41 | 8.61 |
| DoubleTake ($M_{soft}=0.0$) | 0.59 | 0.85 | 9.75 | 5.70 | 1.72 | 8.67 |
| DoubleTake ($M_{soft}=0.2$) | 0.59 | 0.90 | 9.69 | 5.66 | 1.50 | 8.77 |

| | | R-precision↑ | FID↓ | Diversity→ |
|---|---|---|---|---|
| | Ground Truth | 0.80 | $1.6 \cdot 10^{-3}$ | 9.33 |
| **Trajectory** | MDM | 0.63 | 0.98 | 9.04 |
| | Fine-tuned (Ours) | 0.64 | 0.54 | 9.16 |
| **Left Wrist** | MDM | 0.63 | 0.82 | **9.31** |
| | Fine-tuned (Ours) | 0.64 | 0.34 | 9.41 |
| **Left Wrist + Trajectory** | MDM | 0.65 | 1.18 | 8.81 |
| | DiffusionBlending (Ours) | 0.67 | 0.22 | 9.33 |
| **Left Wrist + Right Foot** | MDM | 0.63 | 0.81 | 8.84 |
| | DiffusionBlending (Ours) | 0.67 | 0.18 | 9.35 |

Table 2: **Quantitative results on the HumanML3D test set.** All methods use the real motion length from the ground truth. '→' means results are better if the metric is closer to the real distribution. We run all the evaluations 10 times. **Bold** indicates best result, underline indicates second best result. R-precision reported is top-3, Div. stands for diversity and M.-Dist for Multi-modal distance.

Table 3: **Joints control with fine-tuned models and DiffusionBlending.** We compare our joints control method with the motion inpainting method suggested by Tevet et al. (2023). We conduct the evaluation on HumanML3D test set. $'+'$ sign represents a blending of two fine-tuned models with our DiffusionBlending method.

Table 2 presents ablations for the DoubleTake hyperparameters over the HumanML3D dataset. We show that our method using DoubleTake, soft masking, and one-second handshake size achieves the best results. Figure 5 shows qualitatively how the second take refines the first take.

| | Root Error [m] | | | Joints Error [m] | | |
|---|---|---|---|---|---|---|
| | $1s$ | $2s$ | $3s$ | $1s$ | $2s$ | $3s$ |
| MRT | 0.28 | 0.49 | 0.62 | **0.32** | 0.53 | 0.65 |
| ComMDM (ours) | **0.24** | **0.38** | **0.43** | **0.32** | **0.44** | 0.49 |
| MDM (no Com) | 0.27 | 0.57 | 0.7 | 0.33 | 0.62 | 0.82 |
| Com only | 0.25 | 0.40 | 0.48 | 0.33 | 0.47 | 0.54 |
| ComMDM - 2layers | **0.24** | **0.38** | **0.43** | **0.32** | **0.44** | 0.49 |
| ComMDM - 4layers | 0.26 | 0.40 | 0.45 | 0.35 | 0.47 | 0.52 |
| ComMDM @ layer6 | 0.26 | 0.40 | **0.43** | 0.32 | 0.45 | **0.48** |
| ComMDM @ layer4 | 0.27 | 0.42 | 0.46 | 0.33 | 0.47 | 0.51 |
| ComMDM @ layer2 | 0.26 | 0.40 | 0.45 | 0.33 | 0.46 | 0.50 |
| ComMDM @ layer0 | **0.24** | 0.40 | 0.45 | **0.32** | 0.47 | 0.51 |

Table 4: **CMU-mocap prefix completion L2 error.** Given a 1-second long prefix, all models predict a 3-second long motion completion. We report the root and joints mean error for the first 1, 2, and 3 seconds. **Bold** indicates best result, underline indicates second best. We introduce two ablation studies, the first is for the number of layers constructing ComMDM (ours is 1), and the second is in which layer of MDM it is placed (ours is in the 8th). Observe that the communication block performs better when placed in higher layers of the transformer and constructed from fewer layers.

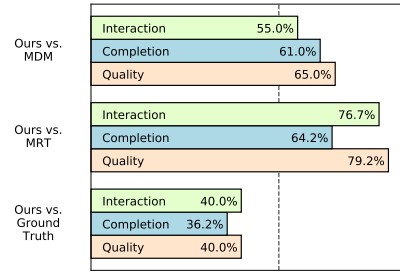

Figure 6: **3DPW two-person prefix completion user study.** We asked users to compare our ComMDM to the original MDM, MRT model, and ground truth in a side-by-side view. The dashed line marks 50%. ComMDM outperforms both MRT and MDM in all three aspects of generation.

## 4.2 TWO-PERSON GENERATION

Due to the limited availability of data, we learn two-person motion in a few-shot manner. We use fixed MDM trained on the HumanML3D dataset and learn a slim communication block, ComMDM, as described in Section 3.

**Data.** We train and evaluate ComMDM with the CMU-Mocap (CMU) and the 3DPW (Von Marcard et al., 2018) datasets, which contain 55 and 27 two-person motion sequences respectively. annotated with SMPL joints. We omit the 3DPW test set since it is noisy and does not include any meaningful human interaction. We augment the data by randomly mirroring and cropping each sequence. Then, we process the data to the HumanML3D joint representation, for compatibility with the original MDM input format. We train ComMDM for two different generation tasks, both with batch size 64 on a single NVIDIA GeForce RTX 2080 Ti GPU.

**Prefix completion.** We follow MRT (Wang et al., 2021) and learn to complete 3 seconds of motion given a 1-second prefix. Table 4 presents the root and joints mean $L2$ error - considering the ablation

study presented, we placed the communication layer in the 8th and last layer of the transformer. We train ComMDM for $240K$ steps. We retrain MRT and observe that our data process alone improved the results originally reported by the authors. ComMDM outperforms MRT in terms of L2 error and generates visibly more meaningful completions, as presented in Figure 7 (Appendix). We further conducted a user study comparing ComMDM to MDM, MRT, and ground truth data, according to the aspects of *interaction level*, *completion of the prefix*, and *overall quality* of the generated motion. 30 unique users were participating in the user study. Each model was compared to ComMDM through 10 randomly sampled prefixes and each such comparison was repeated by 10 unique users. The results (Figure 6) show that the motions generated by ComMDM were clearly preferred over MRT and MDM. Figure 11 (Appeendix) shows an example screenshot from this user study.

**Text-to-Motion.** We argue that prefix completion is a motion task that becomes irrelevant. It is an explicit control signal that is both limiting the motion and giving a too-large hint for the generation. Additionally, reporting joint error promotes dull low-frequency motion and discourages learning the distribution of motion given a condition. Hence, we make a first step toward text control for two-person motion generation. Since no multi-person dataset is annotated with text, we contribute 5 textual annotations for each training sample of both datasets, and train ComMDM for $100K$ steps. Figures 1 and 8 (Appendix) present diverse motion generation given unseen text prompts. We note that due to the small number of samples, generalization is fairly limited to interactions from the same type seen during training.

### 4.3 FINE-TUNED MOTION CONTROL

We compare our fine-tuned models and the DiffusionBlending sampling method with the original MDM model on various control tasks. For that sake, we sample text and control features according to each task from the HumanML3D test set. Motion is generated with the original MDM model by injecting the control features using the original inpainting method suggested by (Tevet et al., 2023). We then generate motions with the fine-tuned model that was trained for a specific control task, using our proposed inpainting method. All fine-tuned models were initialized with the same original MDM we compare with, and trained with our finetuning method for $80K$ steps, with a batch size of $64$. Note that we consider the trajectory to be the angle of the character on $xz$ plane and its linear velocities in that plane (we don't include the vertical position). In the joint control tasks, we take the relative location of the joint with respect to the root location. For composite tasks such as left wrist+trajectory and left wrist+right foot, we apply DiffusionBlending method on the two corresponding fine-tuned models with equal weights ($s = 0.5$). All control experiments conducted on HumanML3D dataset, with text-conditioning and a classifier-free guidance scale of 2.5. Quantitative results presented in Table 3 and qualitative results demonstrated in Figure 9 (Appendix). As can be seen, fine-tuning MDM is crucial for the control task, and produces high-quality results.

## 5 CONCLUSION

In this paper, we have shown that a motion-based prior can be employed for advanced motion generation and control, using three novel composition methods. We have leveraged the diffusion approach itself for the task, and have shown that it lends itself naturally to composition, enabling new tasks with little to no new data. Conceptually, we argue that the diffusion-based generative model can serve as a prior, or a proxy, to the human motion manifold, and thus the advanced techniques only need to address the integration between the parts being composed, relying on the fact that the generated motion is always projected back to the motion manifold. While promising, this initial approach is still in its infancy, and much can be further investigated. In long-sequence generation, for example, we are still limited to the quality of the initial model and the motion may suffer inconsistencies between distant intervals. In two-person motion generation, ComMDM does well at synchronizing motions between two priors, but only for interactions seen during training, lacking generalization. Based on the single-person synthesis case, we expect this approach as well to scale with larger datasets in the future. Nevertheless, two-person synthesis brings new challenges yet to be addressed. For example, future methods should allow for valid contacts between people. Lastly, we note the proposed techniques are not specifically designed for the motion domain. Hence perhaps the most promising avenue for future work is to adapt the techniques described in this paper to other fields of generation, as well as to investigate additional ways to combine the vast knowledge embedded in pretrained generative models for novel tasks.

ACKNOWLEDGEMENTS

We extend our gratitude to Prof. Michiel Van de Panne for his invaluable guidance, and insightful suggestions, which have significantly enriched the quality and rigor of this paper. We thank Chuan Guo and Nikos Athanasiou for their technical support and useful advice. We thank Sigal Raab, Roy Hachnochi and Rinon Gal for the fruitful discussions. This research was supported in part by the Israel Science Foundation (grants no. 2492/20 and 3441/21), Len Blavatnik and the Blavatnik family foundation, and The Tel Aviv University Innovation Laboratories (TILabs). This work was supported by the Yandex Initiative in Machine Learning.

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

## APPENDIX

## A  ADDITIONAL RESULTS

Figures 8, 7 show additional two-person motion generation results with ComMDM for the tasks of text-to-motion and prefix completion correspondingly. Figure 10 qualitatively compares Our DoubleTake algorithm to TEACH (Athanasiou et al., 2022). Figure 9 demonstrates fine-tuning motion control.

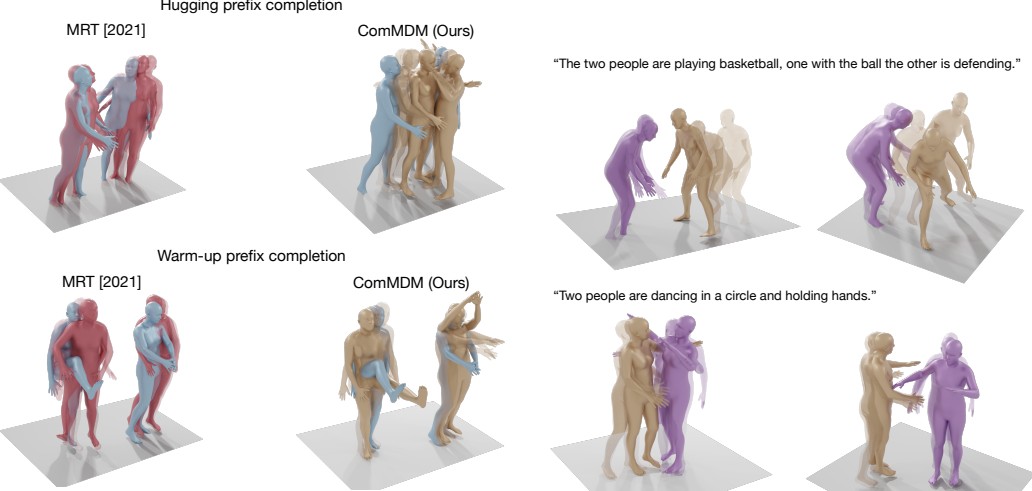

Figure 7: **Two-Person Prefix Completion.** MRT (Wang et al., 2021) tends to fixate on the prefix pose whereas our ComMDM provides lively and semantically correct completions. Blue figures are the input prefix frames, provided to both models. The red and orange figures are MRT and our completions correspondingly.

Figure 8: **Two-Person Text-to-Motion.** We use ComMDM to generate two-person interactions given an unseen text prompt describing it. Different color defines different character, both are generated simultaneously.

## B  FINE-TUNING ALGORITHMS

---

**Algorithm 1** Fine-tuning method

**repeat**
    $x_0 \sim q(x_0)$
    $t \sim \text{Uniform}(\{1, \ldots, T\})$
    $\epsilon \sim \mathcal{N}(0, I)$
    $\epsilon[trajectory] = 0$     ▷ **Our addition**
    Take gradient descent step on:
    $\nabla_\theta \| x_0 - \epsilon_\theta \left( \sqrt{\bar{\alpha}_t} x_0 + \sqrt{1 - \bar{\alpha}_t} \epsilon, t \right) \|$
**until** Converged

---

**Algorithm 2** Sampling method

$x_0^{(T)} = 0$
**for** $t = T, \ldots, 0$ **do**
    $x_0^{(t)}[trajectory] = $given trajectory     ▷
**Original in-painting**
    $\epsilon \sim \mathcal{N}(0, I)$
    $\epsilon[trajectory] = 0$     ▷ **Our addition**
    $x_0^{(t-1)} = \epsilon_\theta \left( \sqrt{\bar{\alpha}_t} x_0 + \sqrt{1 - \bar{\alpha}_t} \epsilon, t \right)$
**end for**

---

## C  USER STUDIES

We conducted a user study of the two-person prefix completion task. Its details can be found in Section 4.2 and the results are presented in Figure 6. Figure 11 presents a sample screenshot from the user study form.

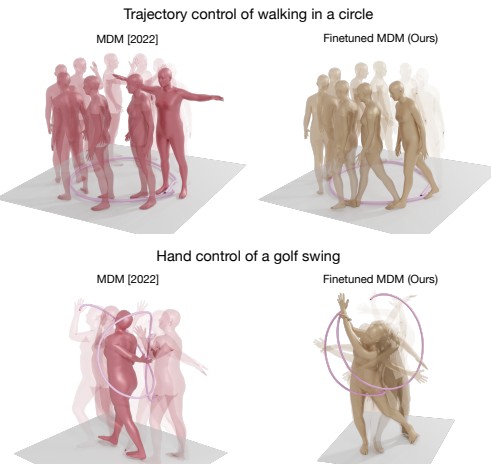
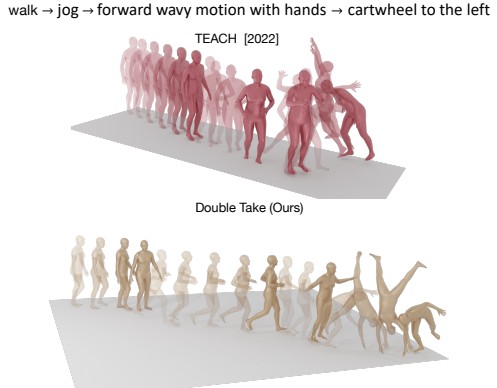

Figure 9: **Fine-tuned Motion Control (unconditioned on text)**. We can see that MDM (Tevet et al., 2023) generates motions that completely ignore the input features: In trajectory control - MDM generates massive foot sliding, and in the hand control, the hand unrealistically bends behind the back. Our finetuned models generate natural motions that semantically and physically match the input features: In trajectory control - we generate a walking motion that fits the trajectory and in hand control, the model recognizes the swinging motion and generates a golf swing.

Figure 10: **DoubleTake compared to TEACH (Athanasiou et al., 2022).** We show that while our DoubleTake provides coherent motion with realistic transitions, TEACH generation suffers from sliding.

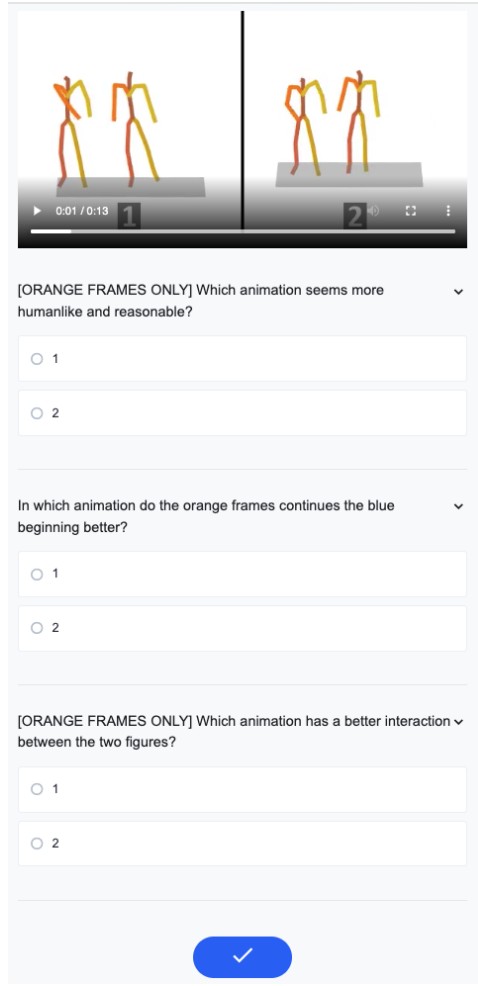

Figure 11: A sample screenshot from the two-person prefix completion user study.

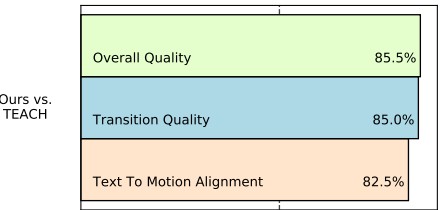

Figure 12: **Long sequences user study.** We asked the users to compare our DoubleTake to TEACH(Athanasiou et al., 2022). The dashed line marks 50%. DoubleTake outperforms TEACH in all three aspects of generation.

In addition, we conducted a user study of the Long Sequences Generation task comparing Double-Take to TEACH (Athanasiou et al., 2022), according to the aspects of (1) alignment with the text condition, (2) natural transitions, and (3) overall quality of the generated motion. 20 unique users were participating in the user study. The motions were generated randomly, sampling texts out of the BABEL dataset. Each comparison was repeated by 10 unique users. The results (Figure 12) show that our long sequences generated were clearly preferred by the users. Figure 13 presents a sample screenshot from the user study form.

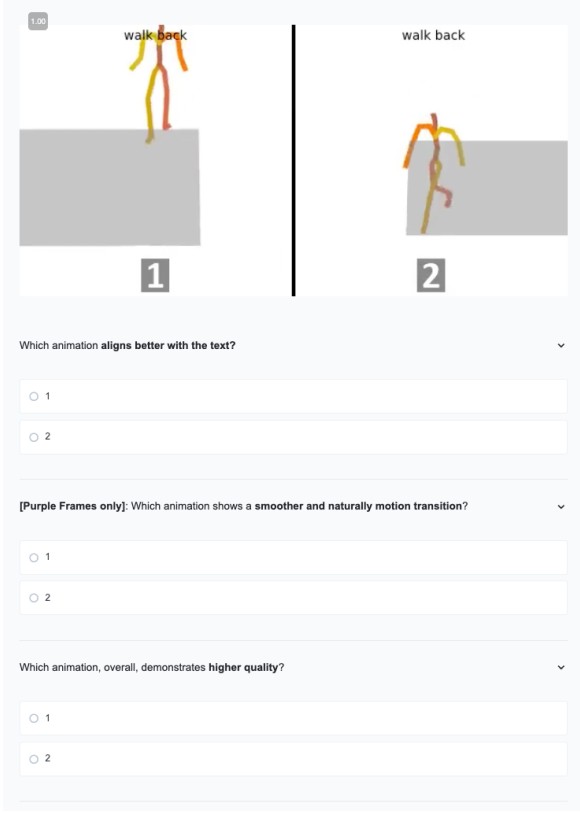

Figure 13: A sample screenshot from the Long Sequence generation user study.

## D  ADDITIONAL PARAMETERS

The followings Table 5 compares DoubleTake and TEACH (Athanasiou et al., 2022) number of parameters and runtime.

| Attribute | DoubleTake | TEACH |
|---|---|---|
| Trainable Parameters | 17.75M | 28.84M |
| Inference time (10 seconds motion) | 78 [s] | 5.7 [s] |
| Inference time (5 minutes motion) | 15.7 [m] | 2.5 [m] |

Table 5: Attributes comparison between DoubleTake and TEACH

