# OpenReview forum: "Human Motion Diffusion as a Generative Prior"
_ICLR.cc/2024/Conference — ICLR 2024 poster_

### Official Review · Reviewer_1uD6 · 2023-10-31

**Soundness:** 3 good
**Presentation:** 3 good
**Contribution:** 2 fair
**Rating:** 6
**Confidence:** 4

**Summary:**

This paper tackle the problem of motion generation. Given a pretrained diffusion based text-to-motion model i.e., MDM, this paper introduce three different way of adaptation to account for long sequence generation, multi-person motion generation and joint trajectory control.  For long sequence generation, two sequences are first generated individually and then blended with a fixed length overlapping. For two-person motion generation, a communication module (ComMDM) is trained to modify the intermediate features from two individual pre-trained MDM so as to coordinate the interaction between them. For trajectory control, several MDMs are frist finetuned to be given the trajectory of different joints. To achieve the control of multiple joints, the paper proposes to interpolate between two MDMs trained beforehand.

**Strengths:**

- The motivation of limited long sequence data, multi-person interaction and detailed control is valid and convincing.
- The proposed method is sound and interesting.
- The experiments are comprehensive and the results on all three tasks are very good especially for the qualitative results.

**Weaknesses:**

- Despite the good results, technically, there are not much contributions. The main contribution is the way of using/finetuning a pre-trained motion diffusion model for three different tasks. However, in each of those tasks the adaptation is straightforward. They are either engineering ticks such as soft or hard masking or  the use of existing techniques such as the interpolation mechanism from Ho & Salimans, (2022).

- The proposed ComMDM is constrained to two persons only. It is hard to scale to multiple persons.

- It is also unclear how the DiffusionBlending can be used to blend more than 2 joints.

**Questions:**

For the DiffusionBlending, is the interpolation applied to the final motion or the intermediate feature as the CommMDM?

---

> ### Author Response · Authors · 2023-11-22
>
> * Regarding a generalization of ComMDM for more than two people - that’s an interesting point. Although presented for the two-person case, ComMDM can get as input the activations of more than two people and output the correction terms for each. In the absence of such a dataset with meaningful interactions, we were not able to validate the performance of ComMDM in such a setting.
> * Regarding DiffusionBlending with more than two joints - following your comment, we added a generalized equation for N joints (in blue). Our initial experiments show that it works well for N=3 and we are willing to add it to the next revision of the paper upon request.
> * Regarding your question about DiffusionBlending - the interpolation is applied to the final output of the model (X0 prediction), similarly to the Classifier-Free-Guidance scheme.

---

> > ### Comment · Reviewer_1uD6 · 2023-11-23
> > **Thank you for the rebuttal.**
> >
> > The rebuttal clarifies some of the details but the concern of lack of technical contribution still remains. I have no further questions and will keep my original rating.

---

### Official Review · Reviewer_xoQq · 2023-11-01

**Soundness:** 3 good
**Presentation:** 3 good
**Contribution:** 3 good
**Rating:** 6
**Confidence:** 5

**Summary:**

This paper leverages a pretrained Motion Diffusion Model (MDM) as a generative prior and introduces three distinct techniques to enhance generation quality and enable new tasks: 1) DoubleTake is employed to reduce artifacts in long-range motion generation, enhancing the smoothness of transitions. 2) The introduction of ComMDM, a compact model integrated after the transformer layer in MDM, allows the model to handle two-person generation by freezing the original MDM and training this new module on a smaller data collection. 3) The authors also propose a fine-tuning method and DiffusionBlending to enhance controllability.

**Strengths:**

1. The paper presents compelling quantitative and qualitative results, setting a new state-of-the-art benchmark with a significant lead.

2. This article broadens the scope of existing text-driven motion generation from three perspectives. The conclusions and experiments associated with these extensions are valuable contributions to the research community.

3. The paper is well written, ensuring that its content is readily comprehensible to its readers.

**Weaknesses:**

1. The authors should conduct a user study to quantitatively compare the visual results of TEACH with the proposed method for long sequence generation. Model parameters and inference speed should also be provided for a more comprehensive performance comparison between the two.

2. Dual-person motion generation lacks comparative experiments, for example, with InterGen \[1\]. This paper introduces the InterHuman benchmark, a large-scale dataset for dual-person motion, and provides more comparative references in the article

\[1\] Han Liang, et al. InterGen: Diffusion-based Multi-human Motion Generation under Complex Interactions

**Questions:**

Please kindly refer to the weaknesses mentioned above.

---

> ### Author Response · Authors · 2023-11-22
>
> * Regarding the comparison with TEACH - Following your comment we have conducted a user study as requested and added the results to the appendix (Figures 11, 12). We also added a paragraph regarding the user study in section 4.1 (in blue). As for the user study, we compared our DoubleTake method to TEACH by sampling randomly out of the validation set of BABEL 20 sequences of long motion such that each is ~10 seconds long in total and generated out of 3 different text prompts and durations. Both (DoubleTake vs TEACH) generated the exact same duration of motion when the transitions were marked with purple color and motions with orange. We asked 3 different question per motion sequence:
> (1)  Which animation aligns better with the text?
> (2) [Purple Frames only]: Which animation shows a smoother and natural motion transition?
> (3) Which animation, overall, demonstrates higher quality?
> Testing Overall Quality, Transition quality and Text To Motion alignment. Most of the users (82 to 85%) preferred our method over TEACH. Each survey had 10 different questions and each question was answered by 10 different users; In total, 20 people answered the survey.
> * Following your request we added the number of parameters and runtime of both TEACH and DoubleTake Appendix, section D (in blue).
> * InterGen follows and extends the two instances’ communication principle presented in ComMDM. As you mentioned, they introduced a new benchmark and used it to evaluate ComMDM as well as their model. Following your comment, we acknowledge their contribution in our related work section (in blue), yet, we will not duplicate their experiment which was already made public.

---

### Official Review · Reviewer_ijG5 · 2023-11-01

**Soundness:** 3 good
**Presentation:** 3 good
**Contribution:** 3 good
**Rating:** 6
**Confidence:** 3

**Summary:**

In this paper, the authors address the challenges in human motion generation by introducing three new composition methods that use diffusion based generative models. The authors align these methods to 3 main challenges: long sequence generation, multi-person interactions, and controllable generation. The authors highlight that much of these problems arise from the lack of available data. The 3 proposed methods are sequential composition for long sequence generation, parallel composition for two-person motion generation, and model composition for fine-grained control and editing. The models involved in these methodologies are respectively DoubleTake for generating long motion sequences in a zero-shot manner, ComMDM to combine two frozen priors to enable two-person motion generation, and DiffusionBlending for flexible control of generated motion. The authors present several qualitative and quantitative results demonstrating the positive effects of their method.

**Strengths:**

I want to highlight the following strengths:
- The main strength I see is that the methods presented work well without the need of generating more data (or consuming large amounts of unavailable data). This is a strong benefit, since the field of human motion generation is still lagging in terms of data availability. The authors demonstrate in all 3 cases that they can satisfy the task at hand requiring small amounts of extra data/training.
- I see major novelty in the methods developed for long sequence generation and 2 person generation. Both methods have interesting new ways to combine different generations from diffusion models (one over time and one in space). Adding to it that the method doesn't require a lot of extra training, these proposed methods seem solid and novel to me.
- The paper is well written, with good experiments on all fronts. The author's explanation of each method is easy to follow. I want to particularly highlight figures 3 and 4, where the choice of colors and graphics makes it very intuitive to understand.

**Weaknesses:**

My only concern is on the fine-tuned motion control part. The task seems very similar to controlled motion generation. In that case, there is a body of literature in this subject, many of which uses diffusion models for controlled motion generation. The authors failed to include these methods and compare against them. Of course these methods have different data requirements, but they seem to achieve the same goal. I put a list of these methods below. I ask the authors to explain why they did not include these methods in their comparisons? I'm still happy with the paper and I think the authors could make a case while still including these works, but I would like to hear from the authors on these choices.

- Jiaxi Jiang, Paul Streli, Huajian Qiu, Andreas Fender, Larissa Laich, Patrick Snape, and Christian Holz. Avatarposer: Articulated full-body pose tracking from sparse motion sensing. ECCV, 2022.
- Du, Y., Kips, R., Pumarola, A., Starke, S., Thabet, A., & Sanakoyeu, A. (2023). Avatars grow legs: Generating smooth human motion from sparse tracking inputs with diffusion model. In Proceedings of the IEEE/CVF Conference on Computer Vision and Pattern Recognition (pp. 481-490).
- Castillo, Angela, et al. "BoDiffusion: Diffusing Sparse Observations for Full-Body Human Motion Synthesis."

**Questions:**

See weaknesses

---

> ### Author Response · Authors · 2023-11-22
>
> * Regarding motion control prior work. This is a valid point, the works you mentioned are closely related to our fine-tuning task, hence, we added a paragraph presenting and discussing them in the related work (in blue). While related, we find an important difference between the two lines of work that leads to two different evaluation settings: Jiang et al., Du et al. and Castillo et al. are solving the task of reconstructing full body motion given the motion of a VR headset and two hand controllers. As such, they are measured with reconstruction metrics (i.e. MPJPE and jitter). On the other hand, our fine-tuned and DiffusionBlending models are aimed at modeling the distribution of motion given one or more control signals - for example, given the left hand’s motion we aim to model the distribution of the right hand (and the rest of the body), not to reconstruct the original motion. Another example is given a trajectory, a person can walk/dance/run, etc. through it. In addition, we enable combining textual conditions to vary this distribution. Hence our evaluation is distribution-based (i.e. FID and diversity). As a result, we think that comparing the reconstruction works in the distribution setting (and vice versa) will be unfair.

---

### Official Review · Reviewer_w1k5 · 2023-11-02

**Soundness:** 4 excellent
**Presentation:** 3 good
**Contribution:** 3 good
**Rating:** 6
**Confidence:** 4

**Summary:**

This paper presents off-the-shelf motion diffusion models as a priori for realizing three forms of synthesis tasks. doubleTake is used to generate long-term human motion. comMDM is used to generate two-person motion. DiffusionBlending is used to enable flexible and efficient fine-grained joint and trajectory level control and editing. Experimental results show that these low-cost composite methods generalize well-trained motion prior to different tasks and outperform previous specialized techniques in the respective tasks.

**Strengths:**

Overall, the paper is well-written with a clear and well-motivated introduction.

The proposed method outperforms previous specialized techniques in the respective task. The experimental designs are comprehensive and show visually appealing results.

**Weaknesses:**

1. For the generation of long sequences, I don't think it makes sense to essentially generate each interval completely independently. A better option would be to use an autoregressive generation method similar to TEACH, but with the smarter option of combining each subsequence. Would it be possible to compare this with a scheme similar to EDGE [1]? Also, the paper admits that a comparison with DiffCollage is not possible due to a lack of publicly available code resources. Their implementation is simple. This may be optional, but would further support the paper.

2. Regarding multiplayer motion generation, considering that MRT is an old paper, it could be considered to include a discussion and comparison with [1,2,3]. MRT focuses on deterministic prediction, and for the evaluation, is the prediction from ComMDM sampled once? A better comparison would be to use [2] as a baseline, to check the performance of diverse prediction and to measure diversity.

[1] Tseng et al. EDGE. Edge: Editable dance generation from music. CVPR 2023

[2] Xu et al. Stochastic Multi-Person 3D Motion Forecasting. ICLR 2023

[3] Peng et al. Trajectory-Aware Body Interaction Transformer for Multi-Person Pose Forecasting. CVPR 2023

[4] Liang et al. InterGen: Diffusion-based Multi-human Motion Generation under Complex Interactions. Arxiv 2023

**Questions:**

1. The comparison with TEACH is not very fair either. Is it possible to have the same setting with TEACH but using DoubleMDM, e.g. noising the start and end intervals as diffusion inversions and then performing DoubleMDM with additional transition frames? In this case, you can use accuracy measurements instead of current metrics such as FID. This is because current metrics are more of a test of the quality of the movement and whether the movement is consistent with the text. It is also important to determine if the transition is good by measuring smoothness and accuracy.

Overall, the author's response to the concerns is needed to make the final decision. I am also happy to increase the rating if my concerns are addressed.

---

> ### Author Response · Authors · 2023-11-22
>
> * Regarding the long sequence generation - DoubleTake **does not** generate each interval completely independently. At the first take, the transition parts of the motions are shared between the neighboring intervals as demonstrated in Figure 3. The first take is similar to the EDGE scheme, with a major difference - while EDGE suggests complete overlap, in our case, the overlap is only for the transition part, which is more appropriate to our application.
> * Regarding ComMDM - Thank you for referring us to “Stochastic Multi-Person 3D Motion Forecasting”. We added it to the related work (in blue) and will evaluate ComMDM using the metrics presented in the paper. Since the evaluation code seems missing from their published code base, we contacted the authors and will include this evaluation in our next revision.
> * Regarding the method suggested to compare with TEACH - DoubleTake generates long sequences by generating all the intervals simultaneously with their shared transition. We are afraid that generating transitions in a stand-alone manner will divert too much from our suggested method. Yet, we do suggest an evaluation setting that measures the quality of the motion and the quality of transitions independently. Additionally, in our ablation studies, we evaluate the second take by itself in a setting that quite resembles your suggestion. Since the transitions are generated simultaneously with the motion, and not stand-alone given input motions, we cannot measure its accuracy. Instead, we evaluate the quality and playability of the motion using FID and diversity. To address your concern regarding the fairness of the evaluation we conducted an additional user study comparing to TEACH (in blue) which shows a clear advantage to our DoubleTake considering the quality of motion, transition, and alignment with the input text.

---

### Author Response · Authors · 2023-11-22
**General response**

We thank the reviewers for their comprehensive reviews and positive feedback.
We were pleased to hear you thought the paper is well-written (w1k5, ijG5, xoQq), the experiments are good and comprehensive(w1k5, ijG5, xoQq, 1uD6), the qualitative results and visuals as appealing (XoQq, iJG5, w1k5, 1uD6). Additionally, w1k5 mentioned the strengths of our methodology in addressing various challenges in motion generation, and ijG5 mentioned the methods' novelty and efficiency without extensive data are encouraging.

We carefully read your comments, concerns, questions, and suggestions. According to them, we revised the paper **(see changes in blue)**. In addition, we did our best to address each of them in the following individual responses.

---

### Meta-Review · Area_Chair_YdWN · 2023-12-10

**Metareview:**

The paper received 4 reviews, which placed it slightly above the borderline. The reviewers highlighted that the paper is well written and features superior performance to prior works. The authors got several questions about multiperson generation, which they addressed well as far as AC can tell. Hence the decision is to accept the manuscript.

**Justification For Why Not Higher Score:**

The AC doesn't think the score should go higher as there is not enough support from reviewers.

**Justification For Why Not Lower Score:**

The score shouldn't go lower as all four reviewers are in favor of acceptance.

---

### Decision · Program_Chairs · 2024-01-16

Accept (poster)